# Evolution of prosocial punishment in unstructured and structured populations and in the presence of antisocial punishment

**Mohammad Salahshour**(ORCID)*

Max Planck Institute for Mathematics in the Sciences, Leipzig, Germany

* mohammad.salahshour@mis.mpg.de

**Data Availability Statement:** All relevant data are within the paper and its Supporting information files.

## Abstract

A large body of empirical evidence suggests that altruistic punishment abounds in human societies. Based on such evidence, it is suggested that punishment serves an important role in promoting cooperation in humans and possibly other species. However, as punishment is costly, its evolution is subject to the same problem that it tries to address. To suppress this so-called second-order free-rider problem, known theoretical models on the evolution of punishment resort to one of the few established mechanisms for the evolution of cooperation. This leaves the question of whether altruistic punishment can evolve and give rise to the evolution of cooperation in the absence of such auxiliary cooperation-favoring mechanisms unaddressed. Here, by considering a population of individuals who play a public goods game, followed by a public punishing game, introduced here, we show that altruistic punishment indeed evolves and promotes cooperation in the absence of a cooperation-favoring mechanism. In our model, the punishment pool is considered a public resource whose resources are used for punishment. We show that the evolution of a punishing institution is facilitated when resources in the punishment pool, instead of being wasted, are used to reward punishers when there is nobody to punish. Besides, we show that higher returns to the public resource or punishment pool facilitate the evolution of prosocial instead of antisocial punishment. We also show that an optimal cost of investment in the punishment pool facilitates the evolution of prosocial punishment. Finally, our analysis shows that being close to a physical phase transition facilitates the evolution of altruistic punishment.

## Introduction

Cooperation requires a cooperator to incur a cost for others to benefit. As such, cooperation is costly and expected to diminish by natural selection [1, 2]. Contrary to this rational expectation, cooperation is everywhere-present in the biological and social world [3–6]. Empirical studies suggest, by enforcing cooperation in animal [7–10] and human societies [6, 11–15], altruistic punishment can play an important role in the evolution of cooperation. However, just as cooperation does, altruistic punishment, being costly, goes against an individual's self-interest, and its evolution is yet another puzzle [16, 17]. Many studies have tried to address this

**Funding:** The author acknowledges funding from Alexander von Humboldt Foundation in the framework of the Sofja Kovalevskaja Award endowed by the German Federal Ministry of Education and Research. The funders had no role in study design, data collection and analysis, decision to publish, or preparation of the manuscript.

**Competing interests:** The authors have declared that no competing interests exist.

so-called second-order free-rider problem, according to which free-riding on prosocial punishers who punish defectors results in the elimination of punishers, and subsequently, to the extinction of cooperators due to first-order free-riding on cooperators. Theoretical models have shown that these two problems, first-order and second-order free-rider problems, can simultaneously be solved in cooperation favoring environments; that is when one of the few known mechanisms for the evolution of cooperation is at work. In this regard, group selection [18], indirect reciprocity [19–21], voluntary participation [22–25], spatial selection and network structure [26–29], prior commitment in binary interactions (i.e., in the context of prisoner's dilemma) [30], stochastically bribing punishers [31], and probabilistic sharing of the cost of punishment [32] have been successfully appealed to show how cooperation and punishment can co-evolve in cooperation favoring environments. Other studies have shown once exogenously established, punishment can have a positive effect on the evolution of cooperation [33]. However, the important question of whether altruistic punishment on its own can promote cooperation has remained unaddressed.

The theoretical grounds appear more disappointing, when it is noticed that in many cooperation favoring environments where it is argued that punishment and cooperation can co-evolve, the inclusion of a complete set of possible strategies, by adding antisocial punishers (who defect and punish cooperators) to the population, leads to the invasion of antisocial punishers, and thus, undermines the co-evolution of prosocial punishment and cooperation [34–36]. A priory, there is no reason why antisocial punishment should be excluded in the model. Instead, especially given the empirical evidence that antisocial punishment is abundant in human and animal societies [37–40], its exclusion is an important point which the theory needs to address, especially if the endogenous evolution of social norms is concerned. Although this problem is solved in some cases of cooperation-favoring environments (for example, in the case of structured populations [28], when a reputation mechanism is at work [20], or when participation is voluntary, and the type of institutions are observable [25]), a satisfactory understanding of the extent to which antisocial punishment can prevent the evolution of prosocial behavior is still lacking. These two problems, the second-order free-rider problem and, to a lesser degree, the antisocial punishment problem, raise important questions about the evolution of altruistic punishment and its role in the evolution of cooperation, not only in a general environment but also in many cooperation-favoring environments.

Here, by considering a well-mixed population of individuals who play a public goods game, followed by a public punishing game, introduced here, we show that the first-order and the second-order free-rider problems can be solved simultaneously in a general environment, i.e., in the absence of any cooperation-favoring mechanism and in the presence of antisocial punishment. This establishes altruistic punishment as a fundamental road to the evolution of cooperation and explains its evolution. Furthermore, by considering the same model in a structured population, we show that the mechanism is further strengthened in a cooperation-favoring environment. Besides, we show that an optimal weight of second-order versus first-order punishment can facilitate the evolution of prosocial punishment. Our analysis also shows that higher returns to investment in the punishment pool facilitate the evolution of prosocial as opposed to antisocial punishment. We also show that while a too low cost of punishment leads to the evolution of antisocial punishment, a too high punishment cost prohibits the evolution of punishing institutions, be it prosocial or antisocial. On the other hand, an optimal cost of investment in the punishment pool, comparable to the cost of investment in the public resource, facilitates the evolution of prosocial punishment. Finally, we provide evidence that being close to a physical phase transition facilitates the evolution of prosocial punishment in structured populations. This parallels some arguments according to which being close to a physical phase transition can facilitates biological functions [41–44].

The public punishing game introduced here can be thought of as a model of pool or institutional punishment. In contrast to peer punishment, where agents individually decide whether to punish others or not, in pool punishment, agents can contribute to erect a punishment institution for conducting punishment [24, 29, 45]. However, in contrast to the previous models of pool punishment [24, 29, 45], in our model, punishment pool is modeled as a public goods game, the resources of which are used for punishment. Besides, while most of the previous models only consider a prosocial punishment pool [24, 29, 45], we allow for the existence of both a prosocial and an antisocial punishment pool. We argue how the public punishing game admits an intuitive interpretation in terms of the law enforcing institutions commonly observed in human societies [46, 47]. In this regard, we argue that increasing the adaptivity of the model in a way that it more closely resembles human punishing institutions can make the co-evolution of cooperation and prosocial punishment possible even in more hostile conditions for the evolution of cooperation. To do so, we show that the evolution of punishing institutions is facilitated if, instead of being wasted, the resources in the punishment pool can be used for rewarding purposes when there is nobody to punish.

## The model

To see how altruistic punishment can evolve and promote cooperation, we consider a population of $N$ individuals in which groups of $g$ individuals are formed at random to play a public goods game (PGG). This game is frequently appealed to in studies on the evolution of cooperation [11–14, 16, 17, 48–50]. In this game, each individual can either cooperate or defect. Cooperators pay a cost $c$ to invest in a public resource. Defectors invest nothing. All the investments are multiplied by an enhancement factor $r$ and are divided equally among all the group members. In addition to playing the public goods game, individuals can engage in prosocial or antisocial punishment. For this purpose, cooperators can invest an amount $c'$ in a prosocial punishment pool. In the same way, defectors can invest the same amount $c'$ in an antisocial punishment pool. All the investments in the prosocial and antisocial punishment pools are multiplied by a punishment enhancement factor $\rho$ and are used for punishment purposes. To this goal, a fraction $1 - \alpha$ of the total resources in the prosocial punishment pool is spent to punish defectors in the group, and a fraction $\alpha$ is used to punish cooperators who do not contribute to the prosocial punishment pool. In the same way, a fraction $1 - \alpha$ of the total resources in the antisocial punishment pool is spent to punish cooperators, while a fraction $\alpha$ is used to punish defectors who do not contribute to the antisocial punishment pool. Thus, $\alpha$ determines the relative strength of second-order with respect to first-order punishment.

Individuals gather payoffs according to the payoff structure of the game and reproduce with a probability proportional to the exponential of their payoff such that the population size remains constant. That is, each individual in the next generation is offspring to an individual in the last generation with a probability proportional to the exponential of its payoff. Offspring inherit the strategies of their parent subject to mutations. We assume mutations in the decisions of the individuals to contribute to the public pool, and their decision to contribute to the punishment pool occurs independently, each with probability $v$. In this study, unless otherwise stated, we set $c = c' = 1$.

We note that the model of punishment, introduced here, has similarities with the models of pool punishment [24, 29]. Pool punishment is introduced in two slightly different ways in the literature. In a popular variant [45], considered, for instance, in ref. [29], punishers invest an amount $B$ to a punishment pool, and free-riders get punished by an amount $G$ provided at least one pool punisher exists in their group. In another variant, considered in ref. [24], punishers invest an amount $B$ in a punishment pool, and free-riders get fined by an amount $G$ per

punisher in their group. In this way, a free-rider gets fined by $Gn_P$, where $n_P$ is the number of punishers in its group. In both variants of pool punishment models discussed above, the amount of resources available for punishment is variable and depends on the number of defectors in the group: It is equal to $n_D G$ in the first variant and $n_D n_P G$ in the second variant, where $n_D$ is the number of defectors in the group. However, it is unclear why the same contribution $B$ to the punishment pool should yield different outcomes (from $G$ to $n_D G$) depending on the number of defectors in the group. Contrary to this feature of the past models, it can be argued that punishment resources should be determined based on the amount of contribution made by punishers to the punishment pool. This is the case in our model, where the total amount of resources available for punishment is determined by the amount of contribution to the punishment pool and is equal to $\beta n_{PC} c'$ (in the case of prosocial punishment pool) or $\beta n_{PD} c'$ (in the case of antisocial punishment pool), where, $n_{PC}$ and $n_{PD}$ are, respectively, the number of prosocial and antisocial punishers in the group. In this regard, the punishment pool is regarded as a public resource, the contributions to which are multiplied by an enhancement factor and are used for punishment purposes.

Looking at the punishment pool as a public resource allows for a modification of the model with interesting consequences. In the model considered so far, if there is nobody to punish, the resources in the punishment pool are wasted. Based on this feature, we call this model the wasteful punishment model. This feature, while keeping the model simple, might not be realistic. In real-world punishing institutions, wealth is not destroyed. Instead, if not necessary for sanctioning purposes, it can be used for other purposes, such as charity or reward. To take this fact into account, we also consider a second model, a non-wasteful punishment model, in which if there is nobody to punish, resources in the punishment pool are divided among its contributors. More precisely, in the non-wasteful punishment model, a fraction $\alpha$ of the resources in the prosocial punishment pool is spent to punish non-punishing cooperators. However, if there is no non-punishing cooperator in the group, instead of being wasted, this is divided equally among the contributors to the prosocial punishment pool (i.e., among the punishing cooperators). Similarly, a fraction $1 - \alpha$ of the resources in the prosocial punishment pool is spent to punish defectors. However, if there is no defector to punish in a group, this fraction is divided equally among the contributors to the prosocial punishment pool. The same holds for the antisocial punishment pool. That is, a fraction $\alpha$ of the resources in the antisocial punishment pool is spent to punish non-punishing defectors. However, if there is no non-punishing defector in the group, this is divided equally among the contributors to the antisocial punishment pool. Similarly, a fraction $1 - \alpha$ of the resources in the antisocial punishment pool is spent to punish cooperators. However, if there is no cooperator to punish in a group, this fraction is divided equally among the contributors to the antisocial punishment pool. All the other details of the model remain the same as before. As we will see, adding this conservation of punishment resources to the model facilitates the evolution of punishing institutions.

## Results

### Wasteful punishment

The phase diagrams of the model in the $r - \rho$ and $r - \alpha$ planes are presented in, respectively, Fig 1(a) and 1(b). The blue circles present the results of simulations in a population of size $N = 10000$, and the red lines result from the numerical solutions of the replicator dynamics, developed in the Methods section. To drive the phase diagram, we have determined the system's equilibrium state starting from a random initial condition in which the strategies of the

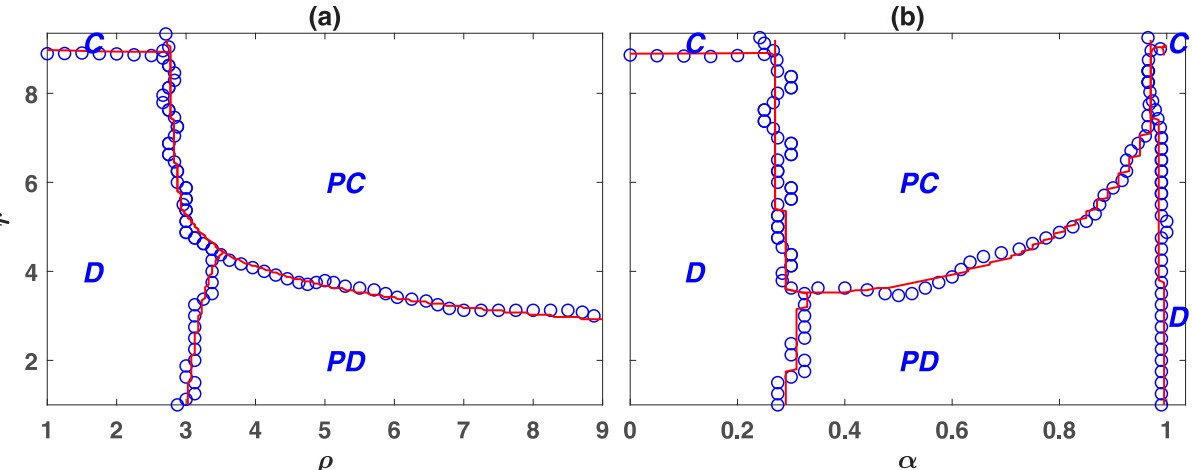

**Fig 1. The phase diagram of the wasteful punishment model in a well-mixed population.** Blue circles denote the results of a simulation in a population of size $N = 10000$, and the red lines denote the results of the replicator dynamics. Depending on the parameters of the model, the model shows four different phases. $C$, $D$, $PC$, and $PD$ denote different phases in which, respectively, cooperators, defectors, punishing cooperators, and punishing defectors dominate. Here, $g = 9$, $v = 0.001$, $c = c' = 1$. In (a) $\alpha = 0.5$ and in (b) $\rho = 5$.

individuals are randomly assigned. For the replicator dynamics, this amounts to an initial condition in which the frequency of all the strategies is the same.

Depending on the model's parameters, the system can be found in one of the four possible phases. As can be seen in Fig 1(a), for small punishment enhancement factors $\rho$, punishing strategies do not evolve. In this region, for $r$ smaller than the group size $g = 9$, the population settles into a defective phase in which only non-punishing defectors survive. $D$ denotes this phase in the figures. As $r$ increases, for a value of $r$ close to $g = 9$, a phase transition to a phase where non-punishing cooperators survive occurs. $C$ denotes this phase. On the other hand, for large values of $\rho$, punishing strategies evolve and eliminate other strategies. However, the nature of the evolving punishment depends on the value of $r$. For small $r$ and large values of $\rho$, such that the return to the investment in the public pool is low, but that to the investment in the punishment pool is high, the population settles into the antisocial punishment phase, where antisocial punishers dominate the population. $PD$ indicates this phase. On the other hand, for large enough values of $r$ and $\rho$, such that the returns to investments in both the public pool and the punishment pool are high, the dynamics settle into the prosocial punishment phase, where prosocial punishers dominate the population. $PC$ denotes this phase.

The phase diagram in the $r - \alpha$ plane shows similar phases. For small $\alpha$, such that there is not enough investment in punishing second-order free-riders, punishment does not evolve. In this region, for $r$ smaller than a value close to $g = 9$ non-punishing defectors survive, and for larger values of $r$ non-punishing cooperators survive. As $\alpha$ increases, a discontinuous transition occurs above which punishing strategies evolve. In this regime, for small $r$ punishing defectors dominate. However, for larger values of $r$, punishing cooperators dominate. For very large values of $\alpha$ (close to 1), another transition occurs above which punishment does not evolve. This shows that enough investment in punishing first-order free-riders is also necessary for the evolution of punishing institutions. Interestingly, for larger values of $\alpha$, the evolution of prosocial punishment requires a larger value of $r$. This shows, an optimal value of $\alpha$, that is, an optimal weight of second-order with respect to first-order punishment exists, which facilitates the evolution of prosocial punishment. Altogether, our analysis reveals that for punishing institutions to evolve, they need to punish both first-order free-riders, who do not

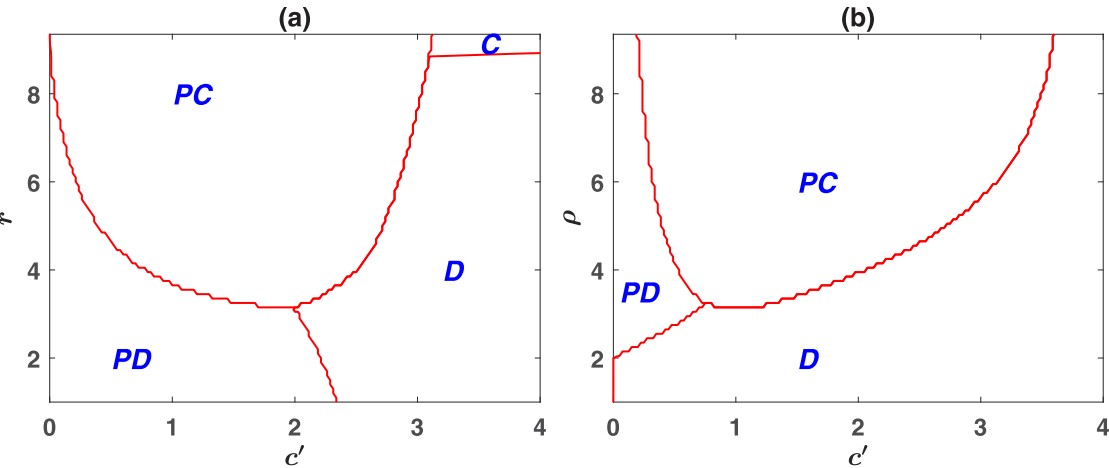

**Fig 2. Dependence on the cost of investment in punishment pool, $c'$.** The phase diagram of the wasteful punishment model in $r - c'$ (a) and $\rho - c'$ planes (b). For too small $c'$ antisocial punishment evolves, and for too large $c'$ non-punishing defectors survive. For medium values of $c'$ prosocial punishment evolves. An optimal $c'$, which is comparable to $c = 1$, facilitates the evolution of prosocial punishment. Parameter values: $g = 9$, $v = 10^{-3}$, $\alpha = 0.5$, and $c = 1$. In (a) $\rho = 5$ and in (b) $r = 5$. The replicator dynamics is used to derive the phase diagrams.

contribute to the public good, and second-order free-riders, who do not contribute to the punishment pool. This suggests that issuing a fine for non-contributors to the policing institutions is necessary for the evolution of such institutions.

An interesting question is how the relative cost of investment in the public pool and the punishment pool affects the evolution of punishment? To address this question, in Fig 2(a) and 2(b), we plot the phase diagram of the model in, respectively, $r - c'$ and $\rho - c'$ planes. As can be seen, for too small cost of punishment, $c'$, antisocial punishment evolves (We note that for $c' = 0$, the fine is zero, and thus, punishment is not possible. The strategies $PD$ and $D$ coexist in this case. A non-zero $c'$ can give rise to a non-zero fine and lead to the domination of antisocial punishers). A non-zero cost of investment in the punishment pool can give rise to the evolution of prosocial punishment. However, the evolution of prosocial punishment is facilitated for medium values of $c'$, comparable to the cost of investment in the public resource, $c = 1$. For too large $c'$, punishing institutions do not evolve. Instead, non-punishing defectors or non-punishing cooperators (for $r$ larger than approximately $g = 9$) dominate the population.

So far, we have considered a well-mixed population. As population structure favors the evolution of cooperation, one might expect that prosocial punishment to evolve in structured populations as well. To see this is indeed the case, we present the phase diagram of the model for a structured population, in Fig 3(a) and 3(b). The phase diagram is derived by performing simulations in a population of $N = 40000$ individuals residing on a $200 \times 200$ first nearest neighbor square lattice with Moore connectivity and periodic boundaries. In a structured population, the model shows similar phases to those that appeared for a well-mixed population. However, two shifts are observable in the position of the phase transitions. First, due to network reciprocity, the $D - C$ transition shifts to smaller enhancement factors. We note that, in the absence of punishing strategies, the $D - C$ transition for the same network structure and size occurs for a larger value of $r$. This shows the beneficial effect of introducing the punishing strategies for the evolution of cooperation, even in the parameter regimes where such punishing strategies do not evolve. This interesting phenomenon results from the synergistic effect of rock-paper-scissor-like dynamics and spatial structure, according to which punishing cooperators facilitate the evolution of non-punishing cooperators by eliminating defectors and

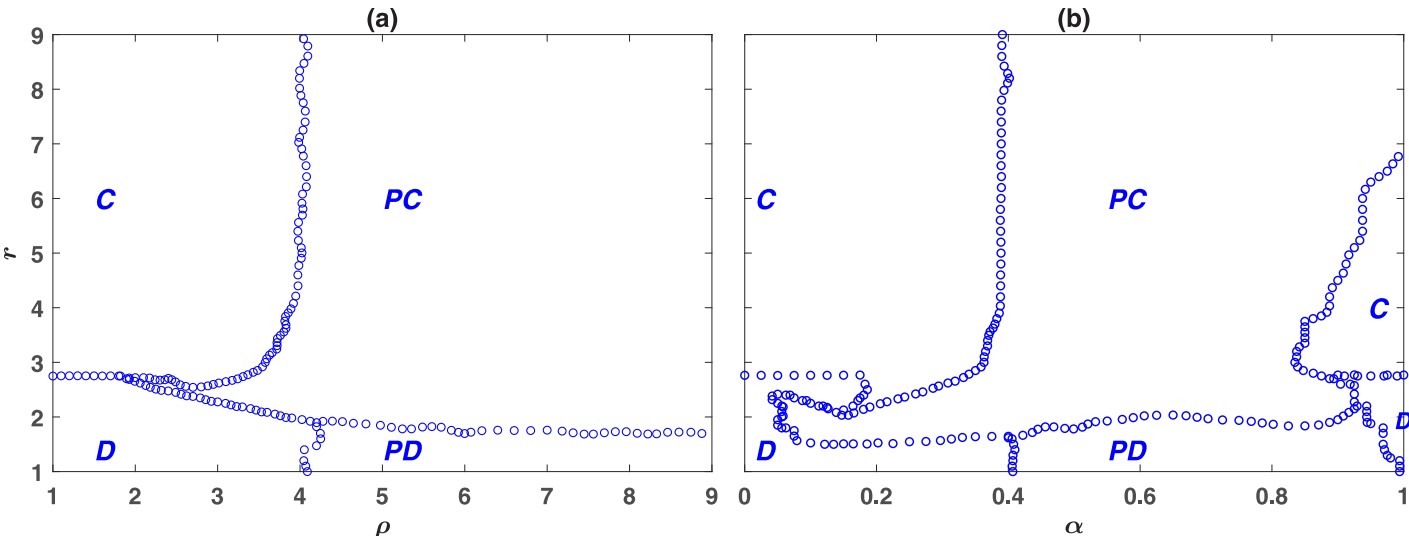

**Fig 3. The phase diagram of the wasteful punishment model in a structured population.** The phase diagram is derived by running simulations in a population of size 40000, residing on a 200 × 200 first nearest neighbor lattice with Moore connectivity and periodic boundaries. Depending on the parameters of the model, the model shows four different phases. *C*, *D*, *PC*, and *PD* denote different phases in which, respectively, cooperators, defectors, punishing cooperators, and punishing defectors dominate. Here, $g = 9$, $v = 0.001$, and $c = c' = 1$. In (a) $\alpha = 0.5$ and in (b) $\rho = 5$.

facilitating the formation of small cooperators block (See S.4.4 and S.5). Second, away from the $D - C$ transition, the phase transition from non-punishing to punishing strategies shifts to larger values of $\rho$, compared to the mixed population. This shows, surprisingly, population structure can hinder the evolution of punishing institutions, be it prosocial or antisocial. On the other hand, the transition from the $PD$ phase to the $PC$ phase occurs for much smaller values of $r$ in a structured population compared to a mixed population. This shows that network structure facilitates the evolution of prosocial as opposed to antisocial punishment. In addition, the analysis of the model reveals that the evolution of prosocial punishment is facilitated close to the $D - C$ transition. This can be observed to be the case by noting that close to the $D - C$ transition, the value of $\rho$ above which prosocial punishment evolves decreases. As shown in the Supplementary Information S1 Text (S.4.2), this result holds for other parameter values and shows the beneficial effect of being close to a continuous transition for the evolution of prosocial behavior. Finally, we note that the $PD - PC$ transition occurs for a smaller value of $r$ compared to the $D - C$ transition. Furthermore, by increasing $\rho$, the $PD - PC$ transition shifts to smaller values of $r$. This is the case in both a well-mixed population and a structured population and shows that the more effective the punishment, the easier, and for smaller enhancement factors, prosocial punishment and cooperation evolve.

To take a closer look at the mechanism by which cooperation and punishment co-evolve, in Fig 4(a)–4(c), we present the snapshots of the time evolution of the system close to the $C - PC$ phase transition. Here, a population of $N = 90000$ individuals residing on a $300 \times 300$ lattice with Moore connectivity and periodic boundaries is considered. The simulation starts with a random assignment of the strategies. The frequency of different strategies as a function of time is plotted in Fig 4(d). Starting from a random initial condition, punishing strategies rapidly grow, while the non-punishing strategies decline. Punishing defectors have the highest growth rate at the beginning of the simulation. This results in a sharp increase in their number by driving other solitary strategies into extinction. However, after small blocks of punishing cooperators are formed, they obtain the highest growth by reaping the benefit of cooperation among themselves and avoiding being punished by rival punishing defectors, and rapidly drive

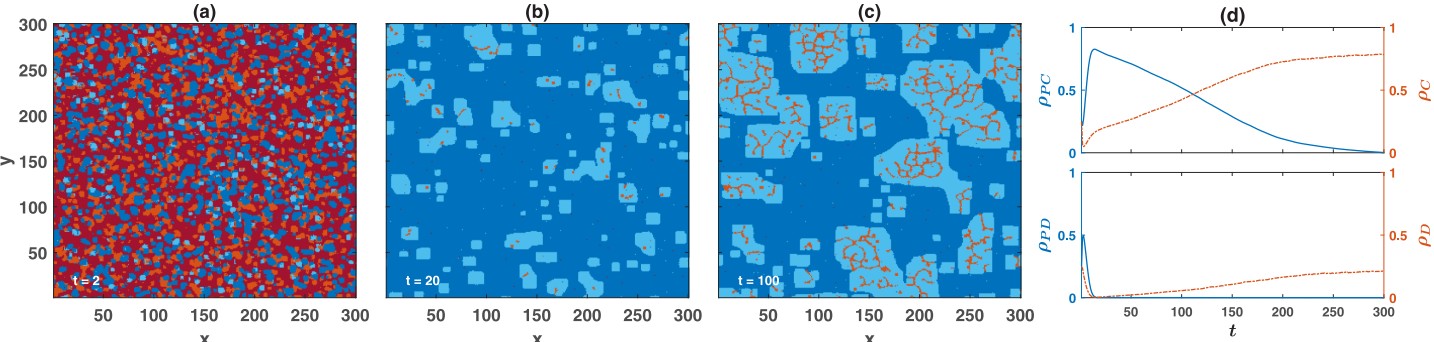

**Fig 4. Time evolution of the wasteful punishment model in a structured population.** (a) to (c) present the snapshots of the system's time evolution, and (d) shows the frequency of different strategies as a function of time. Here, a population of $N = 90000$ individuals lives on a two-dimensional $300 \times 300$ lattice with periodic boundaries and Moore connectivity. Different colors indicate different strategies. Light blue shows non-punishing cooperators, dark blue shows punishing cooperators, light red shows non-punishing defectors, and dark red shows punishing defectors. The simulation started with a random initial condition. Here, $v = 10^{-3}$, $c = c' = 1$, $r = 4$ and $\rho = 3.47$.

punishing and non-punishing defectors into extinction. As argued below, by setting the stage for the invasion of non-punishing cooperators, this phenomenon facilitates the evolution of cooperation. The initial rapid growth of punishing cooperators sets the stage for the second stage of the system's time evolution, in which small domains of non-punishing cooperators are formed in a sea of prosocial punishers. As here, the system is in the $C$ phase, cooperators experience advantage over prosocial punishers. Consequently, cooperators' blocks start to grow slowly along the horizontal and vertical boundaries until they dominate the population. While defectors can not survive in the sea of prosocial punishers, they survive by forming narrow bands within the domain of non-punishing cooperators. Consequently, once non-punishing cooperators start to dominate the population, the frequency of defectors increases as well. We note that, as shown in the S1 Text (S.4.4 and S.5) and the Supplementary Videos (S1–S5 Videos), this coarsening pattern is characteristic of the evolution of punishing strategies.

Finally, we note that in a mixed population, the model is multi-stable in the entire phase diagram; for $r < g$, all the three strategies, $D$, $PD$, and $PC$ are stable. This implies that all the transitions but the $D - C$ transition are discontinuous. As shown in the S1 Text (S.3.2), the nature of the $D - C$ transition depends on the value of $\rho$. While this transition is discontinuous for large $\rho$, for small $\rho$ there is a cross-over from the $D$ phase to the $C$ phase without passing any singularity. In between, the transition becomes a continuous transition at a critical point. Similarly, for a structured population, all the transitions but the $D - C$ transition are discontinuous. The $D - C$ transition, in contrast, shows no discontinuity and appears to occur continuously (S.4.3).

## Non-wasteful punishment

The phase diagram of the non-wasteful punishment model for a mixed population is presented in Fig 5(a). Fig 5(a), presents the phase diagram of the model in the $r - \rho$ plane, and Fig 5(b), presents the phase diagram of the model in the $r - \alpha$ plane. Blue circles represent the result of a simulation in a population of size $N = 40000$, and the red lines represent the result of the replicator dynamics. As can be seen, the result of the replicator dynamics is in good agreement with the result of simulations.

As in the wasteful punishment model, the dynamics in the non-wasteful punishment model is multi-stable: depending on the initial conditions, the dynamics settle into a phase where one of the strategies dominates the population and drives all the other strategies into extinction.

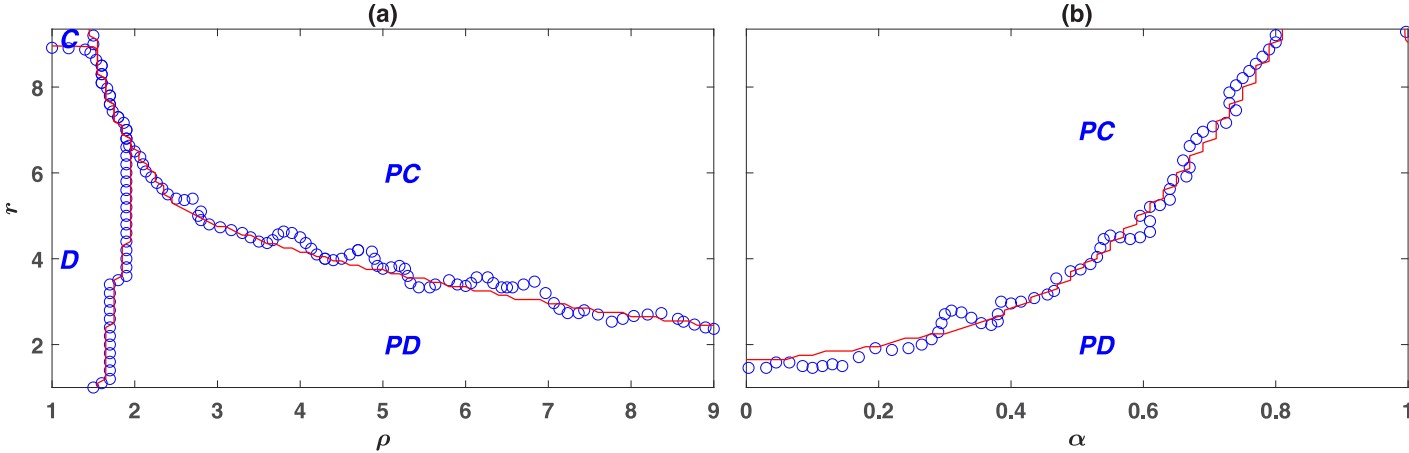

**Fig 5. The phase diagram of non-wasteful punishment model for a mixed population.** Blue circles denote the results of a simulation in a population of size $N = 40000$, and the red lines denote the results of the replicator dynamics. Depending on the parameters of the model, the model shows four different phases separated by discontinuous transitions. *C*, *D*, *PC*, and *PD* denote different phases in which, respectively, non-punishing cooperators, non-punishing defectors, punishing cooperators, and punishing defectors dominate the population. Here, $g = 9$, $v = 0.001$, and $c = c' = 1$.

For small punishing enhancement factors, $\rho$, punishing strategies do not evolve. In this region, for $r$ smaller than a value close to $g$, the dynamics settle into a defective phase in which non-punishing defectors dominate the population. *D* denotes this phase in the figure. On the other hand, for $r$ larger than (approximately) $g$, cooperators survive and dominate the population. *C* denotes this phase in the figure. As $\rho$ increases, a phase transition to a phase where punishing strategies evolve occurs. However, the nature of the evolved punishing strategy depends on the value of $r$. For small $r$ punishing defectors dominate the population. *PD* denotes this phase in the figure. On the other hand, for large values of $r$, the dynamics settle into a phase where punishing cooperators dominate the population. *PC* denotes this phase in the figure.

Comparison with the wasteful punishment model shows that the evolution of punishment is facilitated in the non-wasteful punishment model, be it prosocial or antisocial. This can be seen by noting that the phase transition to the punishing phase occurs for a smaller value of the punishment enhancement factor, $\rho$, in the non-wasteful punishment model. This shows that a smaller return to the investment in the punishment pool is sufficient to give rise to the evolution of punishment, when, instead of being wasted, the resources in the punishment pool are redistributed among its contributors in case there is nobody to punish. We note that similarly to the wasteful punishment model, higher returns to the investments in the punishment pool facilitate the evolution of prosocial as opposed to antisocial punishment. This can be seen by noting that for higher values of $\rho$, the transition to the prosocial punishment phase shifts to smaller values of $r$. That is, for more effective punishment mechanisms, a smaller enhancement factor for the public resource is sufficient to promote social punishment.

The phase diagram of the non-wasteful model in the $\alpha - r$ plane is presented in Fig 5(b). As can be seen, in the non-wasteful model, punishing strategies evolve even for $\alpha = 0$. That is, punishment evolves even in the absence of second-order punishment. The reason is that the prospect of receiving a return from the punishment pool in case there is nobody to punish can act as a reward which solves the second-order free-riding problem. Consequently, second-order punishment is not necessary to ensure the evolution of punishment. Furthermore, the value of $r$ for which prosocial punishment evolves increases by increasing $\alpha$. This shows that second-order punishment can be detrimental to the evolution of prosocial punishment in a

situation where the resources of the punishment pool are rewarded to its contributors when there is nobody to punish.

Finally, we note that similarly to the wasteful punishment model, the non-wasteful punishment model shows multistability in the whole region of the phase diagram: For $r < g$, three stable phases, $D$, $PD$, and $PC$ exist. Depending on the initial conditions, the dynamics settle into one of these phases. On the other hand, for $r > g$, both the $C$ and the $PC$ phases are stable. This implies that the phase transitions involving different punishment phases in this model are discontinuous.

The non-wasteful punishment model in a structured population is studied in the S1 Text and it is shown that occasional rewarding of the resources in the punishment pool can facilitate the evolution of prosocial punishment in a structured population too.

## Discussion

As the evolution of altruistic punishment is riddled by the same kind of free-riding problem that the evolution of cooperation is, it was believed that resorting to another cooperation favoring mechanism is necessary to explain the evolution of altruistic punishment and its role in the evolution of cooperation [18, 19, 21–24, 26–28]. As we have shown, this is not necessarily the case. Instead, the efficient coupling of the second-order and first-order free-rider problems provides a surprising way for the simultaneous solution of both dilemmas. This establishes altruistic punishment as a fundamental road to the evolution of cooperation and can explain its overwhelming presence in human and many animal societies. Furthermore, our study brings new insights into the beneficial conditions for the evolution of punishing institutions and prosocial behavior. In this regard, our analysis shows an optimal weight for second-order, with respect to first-order punishment, exists, which facilitates the evolution of altruistic punishment. Besides, the more efficient the punishment mechanism, the more likely that prosocial punishment, as opposed to antisocial punishment, evolves. Our analysis also reveals that network structure can be detrimental to the evolution of punishing institutions, be it prosocial or antisocial. This theoretical prediction has been observed recently in spatial public goods experiments [51], and parallels some arguments that network structure can sometimes be surprisingly harmful for the evolution of prosocial behavior [52]. On the other hand, network structure can facilitate the evolution of prosocial punishment instead of antisocial punishment, provided favorable conditions for the evolution of punishing institutions are satisfied. We have also seen that an optimal cost of investment in the punishment pool facilitates the evolution of prosocial punishment: While for a too low cost of punishment, antisocial punishment evolves, for too high investment cost punishing institutions, be it prosocial or antisocial, does not evolve.

Finally, we have seen that being close to a physical phase transition is beneficial for the evolution of punishing institutions in a structured population. This parallels many arguments, according to which being close to physical phase transitions can provide optimal conditions for many biological functions and extends such arguments to the evolution of prosocial behavior [41–44].

Past studies have considered the evolution or pool or institutional punishment [24, 29, 45]. The point of departure in our model from the previous models of pool punishment is that the punishment pool is considered a public resource in our model, the resources of which are used for punishing purposes. In other words, in our model, the value of the fines received by punished individuals depends of the investment made by punishers. In addition, while most of the previous models exclude an antisocial punishment pool [24, 29, 45], in our model, an antisocial punishment pool exists as well. Using previous models of pool punishment, past studies

have not detected the evolution of cooperation in a mixed population and in the absence of auxiliary mechanisms. However, as we show by exploring a broad parameter range, the model of pool punishment introduced here can give rise to the evolution of cooperation for sufficiently large values of punishment enhancement factor $\rho$, and proper choice of second-order punishment ratio $\alpha$.

Just as the public goods game is thought of as a metaphor for a social dilemma, the public punishing game introduced here can be thought of as a metaphor for public punishing institutions, such as formal and informal policing institutions at work in human societies [46, 47]. In this regard, the contribution to the public punishing pool can be thought of as a tax paid by individuals to establish a policing institution. Similarly, the punishment of second-order free-riders can be considered as a fine for not paying the tax, and the punishment of first-order free-riders can be considered a fine for not contributing to the public good. Our model can be thought of as a simple and minimal model which grasps the essential aspects of the evolution of such punishing institutions. However, human punishing institutions are adaptive institutions that have accumulated a high level of sophistication in the course of their evolution [46, 47]. In terms of this analogy, the model can be made more adaptive to resemble human sanctioning institutions more closely. Such adaptivity is expected to increase the effectiveness of the punishing institutions and thus, facilitates the evolution of cooperation and prosocial punishment, just as it arguably does in real-world sanctioning institutions. We have considered one such modification in which the resources in the punishing pool, instead of being wasted if they remain unused for punishing purposes, can be used to reward its contributors. As we have seen, such a modification facilitates the evolution of punishment and cooperation. Furthermore, in such a non-wasteful punishment model, second-order punishment is not necessary for the evolution of punishment. Instead, occasional rewarding of punishers can provide an incentive to contribute to the punishing pool, which can solve the second-order free-riding problem. This feature of the non-wasteful punishment model is reminiscent of some arguments according to which combining reward and punishment [33] or second-order reward [53] can be an effective mechanism for promoting social behavior.

## Methods and materials

### The replicator dynamics

In a well-mixed population, it is possible to drive a set of equations for the time evolution of the system in terms of the replicator-mutation equations. In a general case, the replicator-mutation equations can be written as follows:

$$m_x(t+1) = \sum_{x'} v_x^{x'} m_{x'}(t) \frac{w_{x'}(t)}{\sum_{x''} m_{x''}(t) w_{x''}(t)}. \tag{1}$$

Here, $x$, $x'$, and $x''$ refer to the strategies, and can be $PC$, $C$, $PD$, or $D$, referring to, respectively, punishing cooperators, non-punishing cooperators, punishing defectors, and non-punishing defectors. $m_x$ is the frequency of the strategy $x$, $w_x$ is the expected fitness of an individual with strategy $x$, and $v_x^{x'}$ is the mutation rate from the strategy $x'$ to the strategy $x$. Under our assumption that mutations in the strategies of the individuals in the public goods pool, and in the public punishing pool occur independently, these can be written in terms of the probability of mutation $v$, as follows. For those transformations which require no mutations, that is $x = x'$, we have $v_x^{x'} = 1 - 2v + v^2$ (this is the probability that no mutation, neither in the strategy to contribute to the public pool, nor in the strategy to contribute to the punishing pool, occurs). For those rates which require two mutations, one in the decision of the individuals to

contribute to the public pool, and one in their decision to contribute to the punishing pool, we have $v_D^{PC} = v_C^{PD} = v_{PC}^D = v_{PD}^C = v^2$. All the other rates, which require only one mutation, are equal to $v_C^D = v_D^C = v_{PC}^{PD} = v_{PD}^{PC} = v_C^{PC} = v_C^{PD} = v_{PD}^D = v_D^{PD} = v - v^2$.

To use the replicator-mutation equation, Eq (1), we need expressions for the expected fitness of different strategies. These are given by the following equations:

$$
\begin{aligned}
w_{PC} = & \sum_{n_{PD}=0}^{g-1-n_{PC}-n_C} \sum_{n_C=0}^{g-1-n_{PC}} \sum_{n_{PC}=0}^{g-1} \\
& \exp\left[cr\frac{1+n_C+n_{PC}}{g} - (1-\alpha)c'\rho\frac{n_{PD}}{1+n_{PC}+n_C} - c - c'\right]\rho_{PC}^{n_{PC}}\rho_C^{n_C} \\
& \rho_{PD}^{n_{PD}}\rho_D^{g-1-n_{PC}-n_C-n_{PD}}\binom{g-1}{n_{PC},n_C,n_{PD},g-1-n_{PC}-n_C-n_{PD}},
\end{aligned}
$$

$$
\begin{aligned}
w_C = & \sum_{n_{PD}=0}^{g-1-n_{PC}-n_C} \sum_{n_C=0}^{g-1-n_{PC}} \sum_{n_{PC}=0}^{g-1} \\
& \exp\left[cr\frac{1+n_C+n_{PC}}{g} - (1-\alpha)c'\rho\frac{n_{PD}}{1+n_{PC}+n_C} - \alpha c'\rho\frac{n_{PC}}{1+n_C} - c\right]\rho_{PC}^{n_{PC}}\rho_C^{n_C} \\
& \rho_{PD}^{n_{PD}}\rho_D^{g-1-n_{PC}-n_C-n_{PD}}\binom{g-1}{n_{PC},n_C,n_{PD},g-1-n_{PC}-n_C-n_{PD}},
\end{aligned}
$$

$$
\begin{aligned}
w_{PD} = & \sum_{n_{PD}=0}^{g-1-n_{PC}-n_C} \sum_{n_C=0}^{g-1-n_{PC}} \sum_{n_{PC}=0}^{g-1} \\
& \exp\left[cr\frac{n_C+n_{PC}}{g} - (1-\alpha)c'\rho\frac{n_{PC}}{1+n_{PD}+n_D} - c'\right]\rho_{PC}^{n_{PC}}\rho_C^{n_C} \\
& \rho_{PD}^{n_{PD}}\rho_D^{g-1-n_{PC}-n_C-n_{PD}}\binom{g-1}{n_{PC},n_C,n_{PD},g-1-n_{PC}-n_C-n_{PD}},
\end{aligned}
$$

$$
\begin{aligned}
w_D = & \sum_{n_{PD}=0}^{g-1-n_{PC}-n_C} \sum_{n_C=0}^{g-1-n_{PC}} \sum_{n_{PC}=0}^{g-1} \\
& \exp\left[cr\frac{n_C+n_{PC}}{g} - (1-\alpha)c'\rho\frac{n_{PC}}{1+n_{PD}+n_D} - \alpha c'\rho\frac{n_{PD}}{1+n_D}\right]\rho_{PC}^{n_{PC}}\rho_C^{n_C} \\
& \rho_{PD}^{n_{PD}}\rho_D^{g-1-n_{PC}-n_C-n_{PD}}\binom{g-1}{n_{PC},n_C,n_{PD},g-1-n_{PC}-n_C-n_{PD}}.
\end{aligned}
\tag{2}
$$

In the following we explain how these expressions can be derived. In the process, we consider a focal individual in a group where there are $n_{PC}$ punishing cooperators, $n_C$ non-punishing cooperators, $n_{PD}$ punishing defectors, and $n_D = g - 1 - n_{PC} - n_C - n_{PD}$ non-punishing defectors in the group. The term in the large bracket in Eq (2), is the payoff of such a focal individual. In the following, we explain why this is so.

Using the previously mentioned notation for the group composition of a focal individual, $cr\frac{1+n_{PC}+n_C}{g} - c$ is the payoff of a focal punishing or non-punishing cooperator, and $cr\frac{n_{PC}+n_C}{g}$ is the payoff of a focal punishing or non-punishing defector from the public goods game. These are the first terms in the large bracket in Eq (2). A focal punishing or non-punishing cooperator, receives a punishment from the punishing defectors in its group equal to $(1-\alpha)c'\rho\frac{n_{PD}}{1+n_{PC}+n_C}$. This is the second term in the large bracket in the expressions for $w_{PC}$ and $w_C$. In addition, a focal non-punishing cooperator is punished by punishing cooperators in its group, by an amount equal to $\alpha c'\rho\frac{n_{PC}}{1+n_C}$. This is the third term in the large bracket in the expression for $w_C$. In

the same way, a focal punishing or non punishing defector, receives a punishment from the punishing cooperators in its group equal to $(1 - \alpha)c'\rho \frac{n_{PC}}{1+n_{PD}+n_D}$. This is the second term in the large bracket in the expressions for $w_{PD}$ and $w_D$. In addition, a focal non-punishing defector is punished by punishing defectors in its group, by an amount equal to $\alpha c'\rho \frac{n_{PD}}{1+n_D}$. This is the third term in the large bracket in the expression for $w_D$. Finally, as punishing cooperators contribute to both the public pool and the prosocial punishing pool, they pay a cost of $c+c'$. On the other hand, non-punishing cooperators only pay a cost of $c$ to contribute to the public pool. Similarly, punishing defectors, pay a cost of $c'$ to contribute to the antisocial punishing pool. Defectors contribute to none of the pools and pay no cost.

As individuals reproduce with a probability proportional to their payoff, the expected fitness of a strategy can be defined as the expected value of the exponential of the payoff of that strategy. To calculate the expected value of fitness, we note that $\rho_{PC}{}^{n_{PC}} \rho_C{}^{n_C} \rho_{PD}{}^{n_{PD}} \rho_D{}^{g-1-n_{PC}-n_C-n_{PD}} \binom{g-1}{n_{PC},n_C,n_{PD},g-1-n_{PC}-n_C-n_{PD}}$, is the probability that a focal individual finds itself in a group with $n_{PC}$ punishing cooperators, $n_C$ non-punishing cooperators, $n_{PD}$ punishing defectors, and $n_D$ non-punishing defectors. Here, $\binom{g-1}{n_{PC},n_C,n_{PD},g-1-n_{PC}-n_C-n_{PD}} = \frac{(g-1)!}{n_{PC}!,n_C!,n_{PD}!,(g-1-n_{PC}-n_C-n_{PD})!}$ is the multinational coefficient. This is the number of ways that among the $g-1$ group-mates of a focal individual, $n_{PC}$, $n_C$, $n_{PD}$, and $g-1-n_{PC}-n_C-n_{PD}$ individuals are respectively, punishing cooperators, non-punishing cooperators, punishing defectors, and non-punishing defectors. Summation over all the possible configurations gives the expected fitness of different strategies. Using the expressions in Eq (2) for the expected fitness of different strategies in Eq (1), we have a set of four equations which gives an analytical description of the model, in the limit of infinite population size.

As show in the S1 Text (S.2), the replicator dynamics for the non-wasteful punishment model can be derived in the same way.

## The simulations and numerical solutions

Analytical solutions result from numerically solving the replicator dynamics of the model. Simulations of the model are performed according to the model definition. Both simulations and analytical solutions are performed with an initial condition in which all the strategies are found in similar frequencies in the population pool. For the solutions of the replicator dynamics, this is assured by setting the initial frequency of all the four strategies equal to 1/4. For simulations, this is assured by a random assignment of the strategies.

## Supporting information

**S1 Text. Supplemental information text.** An overview of the models is given. The replicator dynamics of the non-wasteful punishment model is derived. The models are further analyzed and the robustness of the results is argued. Supplementary Videos are explained.
(PDF)

**S1 Video. Supplementary video 1.** An example of the time evolution of the wasteful punishment model in the structured population. See S1 Text for a description of video.
(AVI)

**S2 Video. Supplementary video 2.** An example of the time evolution of the wasteful punishment model in the structured population. See S1 Text for a description of video.
(AVI)

**S3 Video. Supplementary video 3.** An example of the time evolution of the wasteful punishment model in the structured population. See S1 Text for a description of video.
(AVI)

**S4 Video. Supplementary video 4.** An example of the time evolution of the wasteful punishment model in the structured population. See S1 Text for a description of video.
(AVI)

**S5 Video. Supplementary video 5.** An example of the time evolution of the wasteful punishment model in the structured population. See S1 Text for a description of video.
(AVI)

## Author Contributions

**Conceptualization:** Mohammad Salahshour.

**Data curation:** Mohammad Salahshour.

**Formal analysis:** Mohammad Salahshour.

**Funding acquisition:** Mohammad Salahshour.

**Investigation:** Mohammad Salahshour.

**Methodology:** Mohammad Salahshour.

**Project administration:** Mohammad Salahshour.

**Resources:** Mohammad Salahshour.

**Software:** Mohammad Salahshour.

**Supervision:** Mohammad Salahshour.

**Validation:** Mohammad Salahshour.

**Visualization:** Mohammad Salahshour.

**Writing – original draft:** Mohammad Salahshour.

**Writing – review & editing:** Mohammad Salahshour.

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
