## [Decision Letter · Decision Letter 0]

7 May 2021

PONE-D-21-10893

The evolution of punishing institutions

PLOS ONE

Dear Dr. Salahshour,

Thank you for submitting your manuscript to PLOS ONE. After careful consideration, we feel that it has merit but does not fully meet PLOS ONE’s publication criteria as it currently stands. Therefore, we invite you to submit a revised version of the manuscript that addresses the points raised during the review process.

The two reviewers have provided constructive and detailed comments. They both agreed that the work is interesting, relevant and would provide a good contribution. However, there are several aspects of the paper that need improvements, for which the reviewers have provided constructive suggestions. Please carefully consider them in the revision of your manuscript.

We look forward to receiving your revised manuscript.

Kind regards,

The Anh Han, Ph.D.

Academic Editor

PLOS ONE

Journal Requirements:

Please include captions for your Supporting Information files at the end of your manuscript, and update any in-text citations to match accordingly. Please see our Supporting Information guidelines for more information: http://journals.plos.org/plosone/s/supporting-information.

Additional Editor Comments:

The three reviewers have provided constructive and detailed comments. They all agreed that the work is interesting, relevant and would provide a good contribution. However, there are several aspects of the paper that need improvements, for which the reviewers have provided constructive suggestions. Please carefully consider them in the revision of your manuscript.

Reviewers' comments:

Reviewer's Responses to Questions

**Comments to the Author**

1. Is the manuscript technically sound, and do the data support the conclusions?

Reviewer #1: Yes

Reviewer #2: Yes

Reviewer #3: Yes

2. Has the statistical analysis been performed appropriately and rigorously? 

Reviewer #1: N/A

Reviewer #2: Yes

Reviewer #3: Yes

3. Have the authors made all data underlying the findings in their manuscript fully available?

Reviewer #1: Yes

Reviewer #2: Yes

Reviewer #3: Yes

4. Is the manuscript presented in an intelligible fashion and written in standard English?

Reviewer #1: Yes

Reviewer #2: Yes

Reviewer #3: Yes

5. Review Comments to the Author

Reviewer #1: Remarks to the Author:

In this manuscript, the author has investigated the coevolution of social punishment and antisocial punishment in the public goods game. By using numerical calculations and simulations, they find that there exists an optimal weight for second-order that facilitates the evolution of cooperation. Furthermore, the author has extended the model to structured populations, and found that the mechanism is further strengthened in a cooperation favoring environment. The motivation of this manuscript is clear and the idea is novel. Thus I recommend the publication of this meaningful manuscript after some minor corrections detailed below:

1. In the Model section, the author uses the term “anti-social punishment”. As a corresponding, I suggest that the author change “social” into “pro-social” (line 61).

2. I suggest that the author change the C, D, PC and PD in Figures 1 and 2 to italics.

3. The author analyzes the replication dynamics by used numerical calculation and simulations, so I suggest the author change the title "The simulations and analytical solutions" to "The simulations and numerical solutions" (line 287).

4. In introduction section, the authors mentioned that altruistic punishment can play an important role in the evolution of cooperation. However, there are still some important literatures have not been reviewed, such as PLoS ONE, 16 (1), e0244592; New Journal of Physics, 16 (8), 083016; Proceedings of the Thirtieth AAAI Conference on Artificial Intelligence. 2016: 2494-2500;.Journal of the Royal Society Interface, 12 (102), 20140935; Mathematical Models and Methods in Applied Sciences, 29 (11), 2127-2149. These literatures are very relevant to the subject of punishment and cooperation.

5. The reference format should be adjusted to meet the requirements of PLoS ONE.

Reviewer #2: The author studied how the punishing institution can evolve with cooperation in the presence of an antisocial punishment strategy. The results showed that punishing cooperators (who investigate both public and punishment pools) can evolve even without cooperation favoring mechanisms when the punishment enhancement factor is high enough and the institution punishes both defectors and the second-order free riders in an appropriate ratio. It implies that punishing both the first-order and the second-order free-riders is essential for the evolution of the punishing institution. The model explanation is clear, and the paper is well organized. However, there are some places to be improved for better readability. So, I recommend the publication of the manuscript in PLOS ONE after a minor revision.

[minor suggestions]

1. In the abstract and introduction, "general environment" sounds unclear. It would be better to describe it specifically (e.g., environments with cooperation favoring mechanisms).

2. In the last sentence of the first paragraph on page 1, the author said "However, the important question that whether altruistic punishment on its own can promote cooperation has remained unaddressed." But I am not sure that the main text can answer this. Did you mention it because the transition from PD to PC happens much lower r-value than the transition from D to C? If so, it would be good to highlight it.

3. Fig. S.4. is explained before Fig. S.3. It would be better to change the order of figures.

Reviewer #3: The manuscript presents a model of evolution of cooperation considering pool punishment. The model considers the evolution of both punisher cooperators and defectors doing antisocial punishment. The main contribution of this model is to show that under the right conditions, cooperation and punishment can evolve without requiring other mechanisms favouring cooperation.

Overall, the paper is well written, the goal and the contribution are well presented. The model used is straightforward, easy to understand and adapted to answer the research question. The results are well presented, with the author using phase transition plots to provide a global view of the results. I am in favour of accepting the manuscript with minor modifications.

My main comment is that the manuscript could be improved by explaining clearly why the model presented obtains cooperation using pool punishment when previous models did not. For instance, in the paper of (Sigmund et al. 2010), they do not observe the evolution of punishment and cooperation when they consider that the participation to pool punishment is compulsory. Yet, they appear to have a similar model than the one presented here. Why does the model presented here find different results? Is it because the author explores a larger parameter space, e.g. high punishment enhancement factor? Or is it because the group size is fixed to a low value of 9? Explaining this difference of results would help the claim of the manuscript to get accepted by a reader, in particular as the novelty of contributions in social evolution are often debated (Kay, Keller, and Lehmann 2020).

On the same note, the authors also claims that it introduces a “public punishing game” where individuals can pay a fee to an institution that punish both free-riders not contributing to the public good game and free-riders not contributing to the punishment pool. However, I struggle to see the difference between this “public punishing game” and pool-punishment systems introduced in previous models , such as in (Perc 2012; Sigmund et al. 2010). If there are no fundamental differences, the author needs to cite these previous work. If there are differences, the author should make them clear in the manuscript.

The supplementary material is very long and I am not sure it is necessary to include all these information (I have read it but I did not proofread it in details). It is not very clear to me the interest of characterising the transitions and I do not know of a community with an interest on this particular topic. If I am not mistaken, the author could cut down the supplementary materials.

A list of minor comments:

• I recommend changing the title because institutions are not truly evolving in this model.

• Abstract: “Nevertheless, theoretical work has been unable to show how this is possible.” . This is not exactly true (as the author explains after) and it downplays a lot of theoretical work. I suggest removing it.

• L29: Remove “its exclusion is a point which a proper theory needs to address.”. This sounds like previous theory were not “proper”.

• L62: What would happen if the amount invested in social punishment can differ from the amount invested in antisocial punishment?

• Add a sentence in the model definition that says that the extended description of the method can be found in supplementary material.

• I think the most common term to describe the proportion of individual with a given strategy is frequency instead of density.

• L125: Why do the simulations not always use the same population size? There are results with population size of 10 000, 40 000 and 90 000.

• L137: I am not sure this result comes (exclusively) from the rock-paper-scissor dynamic. Are not punishers rare for this transition because the punishment enhancement factor is low? Is this result not coming from the spatial structure which is known to favour the evolution of cooperation?

• L180: “This implies that all the transitions but the D - C transition are discontinuous.” What does a discontinuous transition mean exactly?

• L201: “It also reveals network structure can be detrimental to the evolution of punishing institutions.” Is it truly the case? Yes, the punishment enhancement factor needs to be slightly higher, but the public good enhancement factor can be much lower (2 instead of 9).

• L205: “Finally, we have seen that being close to a physical phase transition is beneficial for the evolution of punishing institutions.” Is this the case only in the spatial model? If yes, specify it.

• L211: “public punishing game introduced here,” Again, it is not clear in which way the public punishing game of the manuscript is new. It looks similar to previous pool punishment models.

• The model presented in (Perc 2012) is very similar to the spatial model in this manuscript. Could they be quickly compared in the discussion?

• An important parameter in the model is the punishment enhancement factor and the range of value considered can be high (up to 9). Is there a reason to think that such value would exist in real world? If yes, could the author provide an example?

• Supplementary materials S3.1 L4: a “e” is missing in “th”

Bibrliography:

Kay, Tomas, Laurent Keller, and Laurent Lehmann. 2020. “The Evolution of Altruism and the Serial Rediscovery of the Role of Relatedness.” Proceedings of the National Academy of Sciences, 202013596.

Perc, Matjaž. 2012. “Sustainable Institutionalized Punishment Requires Elimination of Second-Order Free-Riders.” Scientific Reports.

Sigmund, Karl, Hannelore De Silva, Arne Traulsen, and Christoph Hauert. 2010. “Social Learning Promotes Institutions for Governing the Commons.” Nature 466 (7308): 861–63.

6. PLOS authors have the option to publish the peer review history of their article (what does this mean?). If published, this will include your full peer review and any attached files.

Reviewer #1: No

Reviewer #2: No

Reviewer #3: No

---

## [Author Response · Author response to Decision Letter 0]

21 Jun 2021

All the reviewer's suggestions are applied in the revised version. A point-by-point response to reviewers is submitted as a separate file, where changes in the manuscript is addressed or responses to their comments is provided.

---

## [Decision Letter · Decision Letter 1]

6 Jul 2021

Evolution of prosocial punishment in unstructured and structured populations and in the presence of antisocial punishment

PONE-D-21-10893R1

Dear Dr. Salahshour,

We’re pleased to inform you that your manuscript has been judged scientifically suitable for publication and will be formally accepted for publication once it meets all outstanding technical requirements.

Kind regards,

The Anh Han, Ph.D.

Academic Editor

PLOS ONE

Additional Editor Comments (optional):

All comments from reviewers have been satisfactorily addressed. Only a few minor things that can be easily corrected during a later stage.

Reviewers' comments:

Reviewer's Responses to Questions

**Comments to the Author**

1. If the authors have adequately addressed your comments raised in a previous round of review and you feel that this manuscript is now acceptable for publication, you may indicate that here to bypass the “Comments to the Author” section, enter your conflict of interest statement in the “Confidential to Editor” section, and submit your "Accept" recommendation.

Reviewer #1: All comments have been addressed

Reviewer #2: All comments have been addressed

Reviewer #3: (No Response)

2. Is the manuscript technically sound, and do the data support the conclusions?

Reviewer #1: Yes

Reviewer #2: Yes

Reviewer #3: Yes

3. Has the statistical analysis been performed appropriately and rigorously? 

Reviewer #1: Yes

Reviewer #2: Yes

Reviewer #3: Yes

4. Have the authors made all data underlying the findings in their manuscript fully available?

Reviewer #1: Yes

Reviewer #2: Yes

Reviewer #3: Yes

5. Is the manuscript presented in an intelligible fashion and written in standard English?

Reviewer #1: Yes

Reviewer #2: Yes

Reviewer #3: Yes

6. Review Comments to the Author

Reviewer #1: The authors have carefully and thoughtfully revised this paper and it is now suitable for publication.

Reviewer #2: The authors have addressed all comments, and the answers and the revised manuscript are satisfying. Hence, I recommend the publication of the revised manuscript in Plos ONE.

Reviewer #3: I thank very much the author for taking in account my comments and for providing detailed responses to each of them. The author has made the modifications asked or has well justified when he did not. I am for accepting this revised manuscript. Find below minor and facultative comments on the new parts of this revised manuscript.

• I think the new title can still be improved. Based on previous title, the author could go for “The evolution of cooperation with pool-punishment institutions” (the problem I stated in my previous review was that evolution of institutions would consider that institutions change along time in this model while they emerge here). Another alternative is “Institutionalised pool-punishment can lead to the evolution of sustainable cooperation”. Those are just suggestions, and the author is free to keep the title as it is.

• The paragraph L108 to L135 could be simplified. I appreciate that the author gives all these details, but I think that a shorter paragraph could reach the reader more easily. Something along the lines of “Previous model of pool-punishment considered that the value of the fine does not depend of the amount invested by punisher. Contrary to past models, we argued that punishment resources should be based on the amount of contribution made by punishers to the punishment pool. This is the case in our model, where the punishment pool is regarded as a public resource, the contributions to which are multiplied by an enhancement factor and are used for punishment purposes.”

• The lines 130 to 135 would fit better in the discussion. I suggest moving them to L373.

• In Figure 2, the author talks about free punishment, that is c’ = 0. What is the amount of the fine in this case? From the model definition, it should be c’ * rho = 0 so no punishment?

• L371. Add a sentence to explain what the author means by the “punishment pool is considered a public resource”. Something like “In other words, in our model, the value of the fines received by punished individuals depends of the investment made by punishers”.

• The author added reference 7 (Kay et al) following my suggestion. However, I believe there was a misunderstanding. I put this reference in my review to support one of my point but I do not think it fits in the paper. I understand that this was confusing in the review but I suggest to remove it from the references in the manuscript.

7. PLOS authors have the option to publish the peer review history of their article (what does this mean?). If published, this will include your full peer review and any attached files.

Reviewer #1: No

Reviewer #2: No

Reviewer #3: No

---

## [Editor Report · Acceptance letter]

29 Jul 2021

PONE-D-21-10893R1 

Evolution of prosocial punishment in unstructured and structured populations and in the presence of antisocial punishment 

Dear Dr. Salahshour:

I'm pleased to inform you that your manuscript has been deemed suitable for publication in PLOS ONE. Congratulations! Your manuscript is now with our production department. 

Kind regards, 

on behalf of

Dr. The Anh Han 

Academic Editor

PLOS ONE